# Are the Dietary–Nutritional Recommendations Met? Analysis of Intake in Endurance Competitions

**DOI:** 10.3390/nu16020189

**Published:** 2024-01-05

**Authors:** Rubén Jiménez-Alfageme, Javier Álvarez, Arkaitz Garbisu-Hualde, David Romero-García, Daniel Giménez-Monzó, Isabel Sospedra, Eva Ausó, José Miguel Martínez-Sanz

**Affiliations:** 1Physiotherapy Department, Faculty of Health Sciences, University of Gasteiz—EUNEIZ, 01013 Vitoria-Gasteiz, Spain; ruben.jimenez@euneiz.com; 2Estudis de Ciències de la Salut, Universitat Oberta de Catalunya, 08018 Barcelona, Spain; jalvarezgarcia2@uoc.edu; 3Physical Activity and Sport Sciences Departmnet, Faculty of Health Sciences, University of Gasteiz—EUNEIZ, 01013 Vitoria-Gasteiz, Spain; arkaitz.garbisu@euneiz.com; 4Faculty of Health Sciences, University of Alicante, 03690 Alicante, Spain; drg33@gcloud.ua.es; 5Department of Community Nursing, Preventive Medicine and Public Health and History of Science Health, University of Alicante, 03690 Alicante, Spain; dgimenez@ua.es; 6Nursing Department, Faculty of Health Sciences, University of Alicante, 03690 Alicante, Spain; josemiguel.ms@ua.es; 7Department of Optics, Pharmacology and Anatomy, Faculty of Sciences, University of Alicante, 03690 Alicante, Spain

**Keywords:** nutrition, endurance, sport supplements, foods, triathlon, trail running

## Abstract

Background: In recent decades the number of endurance events has increased, as well as the number of athletes participating in them. Adequate nutritional and water planning is essential to maintain optimal sports performance and to reduce the incidence of gastrointestinal problems. The main objective of this study is to determine the dietary intake and compliance with nutritional recommendations of athletes in two endurance competitions, as well as to assess the incidence of gastrointestinal complaints. Methods: An observational and cross-sectional study was carried out on the consumption of liquids, food, and supplements in 42 triathletes and mountain runners (MRs) participating in a Vi-Half-Gasteiz triathlon and the Ultra Sierra de Cazorla trail run. At the completion of the trials, participants completed a validated questionnaire (NIQEC). Results: The mean caloric intake during the test of the participants in this study was 192.17 kcal/h, while the mean carbohydrate intake was 43.67 g/h, the mean sodium intake was 267.43 mg/h, and the mean caffeine intake was 15.53 mg/h, with no significant differences between the two sports. The amount of liquids consumed by the participants was 421.21 mL/h, with no significant differences between the triathletes and MRs. As for gastrointestinal problems, it was observed that the participants presented gastrointestinal discomfort in 61.9% of the cases. Conclusions: The intakes of energy, carbohydrates, water, sodium, and caffeine were lower than the current recommendations. There were no differences in the energy, carbohydrate, water, sodium, and caffeine intakes between the triathletes and mountain runners. Gastrointestinal problems showed a high prevalence in these athletes.

## 1. Introduction

Participation in endurance and ultra-endurance events has experienced great growth in recent decades [1,2]. Among the most practiced is the triathlon, characterized by three sports disciplines: swimming, cycling, and running, which are carried out consecutively, in order to complete the competition in the shortest possible time [3]. This sport offers a wide range of event formats, one of the most popular being the middle distance (1900 m swimming, 90 km cycling, and 21.1 km running) [4]. On the other hand, mountain races (MRs), are characterized by being competitions developed in different environments, from low to medium and high mountain paths with various slopes and technical difficulties, in the shortest possible time and respecting the natural environment. [5]. These can be classified according to the distance covered, finding races from less than 21 km to more than 200 km [5], and ultramarathons can be defined as those covering more than 42.2 km [6].

Carbohydrates (CHO) are the main and most important nutrient for the physical development of the endurance athlete [7,8] because as the sporting event progresses, glycogen levels are progressively reduced, and the amount of fuel available is less [1,7,9]. The type of sport, duration, and intensity will determine the amount of carbohydrates to be consumed during physical activity. For endurance events, lasting more than 2.5–3 h and with a competitive-level intensity, the amount should be 90 g per hour [8,9,10]. It is of special importance to consider also the timing of intake. Regarding CHO, as a filling strategy, the consumption of 7–12 g/kg 24 h prior and 10–12 g/kg 36–48 h prior to the event is recommended in tests longer than 90 min of duration, as well as consuming between 1–4 g/kg between 1 and 4 h prior [11]. The type of CHO will also depend on the duration and the athlete’s preferences [10,12], trying to avoid high-fiber, high-fat, and high-protein options [12,13,14].

After exercise, CHO intake is important to replenish glycogen stores, to contribute to the energy needs of the immune system, and to repair damaged tissues [15]. An intake of 0.8–1 g/kg/h of rapidly absorbed carbohydrates is recommended during the first 4 h post-exercise, continuing the rest of the day until having reached 5–7 g/kg, with the inclusion of proteins (0.2–0.4 g/kg/h or 25–30 g/h) being recommended [16]. Intestinal training is essential to increase the rate of CHO oxidation and to avoid the development of gastrointestinal problems during exercise, allowing carbohydrate oxidation by up to 120 g/h [17]. In addition, the combined intake of different types of carbohydrates increases their oxidation rate and reduces the possibility of presenting digestive problems [2,7,13].

On the other hand, maintaining an adequate state of hydration is essential to maintain optimal physical development [18]. The goal should be to prevent both dehydration and overhydration, which can contribute to the development of exercise-associated hyponatremia (EAH) [19]. During intense physical activity or when weather conditions are adverse, it is advisable to take intakes of 0.6–1 l/h, dividing the intake into smaller intakes of 150–250 mL every 15–20 min and always with an isotonic content [20]. Along with these recommendations, to prevent the aforementioned EAH, sodium should be included; the recommended amount of sodium per hour of competition is between 300–600 mg/h [19]. 

One of the main problems in these sports is gastrointestinal discomfort, including reflux, heartburn, belching, bloating, stomach cramps or pain, nausea, vomiting, or diarrhea [14,21]. The intake of nutrients such as fats, fiber, proteins, or highly concentrated CHO solutions has been associated with an increased occurrence of gastrointestinal problems during physical activity [21,22]. 

Despite the above, previous research, such as the work of Jimenez-Alfageme et al. [23], Martinez et al. [24], or Pfeiffer et al. [22], shows that endurance athletes do not reach the recommended energy intakes or an appropriate water and electrolyte intake. The present study is one of the first to compare dietary–nutritional intakes and the incidence of gastrointestinal problems in different long-term endurance sports. In addition, the previous literature is very limited in these types of descriptions. Therefore, the aim of this study is to evaluate nutrient intake and water consumption, as well as the presence of gastrointestinal problems in triathletes participating in the middle-distance triathlon Vi Half Gasteiz 2022 and in mountain runners participating in the Spanish Cup Ultra Sierra de Cazorla 2022. 

## 2. Materials and Methods

### 2.1. Design 

This is an observational and cross-sectional study on the consumption of nutrients, fluids, and supplements and the occurrence of gastrointestinal discomfort by triathletes participating in the Vi Half Gasteiz 2022 middle-distance triathlon and mountain runners participating in the Ultra Sierra de Cazorla 2022. The sample size was calculated with the Rstudio software (version 3.15.0, Rstudio Inc., Boston, MA, USA). The significance level was set a priori at *p* = 0.05. The standard deviation (SD) was set according to the carbohydrate intake (g/h) data from previous studies on ultra marathon runners (SD = 15.2) [24]. With an estimated error (d) of 4.6, the sample size needed was 42 subjects. The study population was selected by non-probabilistic, non-injury, convenience sampling among organizing institutions of competition. The protocol followed, at all times, the World Medical Association codes and Declaration of Helsinki for research in humans and was approved by the ethics committee of the University of Alicante with the file number UA-2022-02-01. In addition, the study design as well as the development of the manuscript followed the STROBE statement [25].

### 2.2. Participants and Sample Size

There were a total of 42 athletes, 32 triathletes, and 10 mountain runners whose anthropometric and sports characteristics are detailed in Table 1. All participants were of legal age; had participated in regional, national, and/or international competitions for at least 1 year; and had not suffered injury or illness in the previous 6 months to completing the questionnaire. The competitive level was classified as regional (competitions at the provincial and/or autonomous community level), national (competitions throughout Spain), and international (competitions throughout the world). All the athletes included in this study participated voluntarily, giving their prior consent for their participation. Table 1 shows the mean and standard deviation of the descriptive data of the sample, segmented according to the sport they performed.

### 2.3. Instruments

A validated online self-administered NIQEC questionnaire, developed specifically to obtain fluid, food, and supplement intake and to determine the incidence of gastrointestinal complaints in endurance competitions, was used [26]. The questionnaire consists of 50 questions and contains five main sections: (1) sociodemographic data; (2) sports data; (3) food, liquid, and supplement intake in the hour before, during, and in the hour after the competition; (4) possible gastrointestinal complaints; and (5) dietary–nutritional planning of the test. The coding of the variables and the estimation of energy and macronutrients was carried out by a trained dietitian–nutritionist, using the Spanish Database of Food Composition (BEDCA) [27] and the data sheets of each sports supplement consumed by the athletes. The questionnaire can be viewed in the Appendix A.

### 2.4. Procedure

To select the study participants, the Vihalf Gasteiz 2022 triathlon organizers and the national representatives of the Spanish Federation of Mountain and Climbing Sports (FEDME) were contacted by e-mail to inform them of the characteristics of this study and to request their collaboration. The questionnaire was sent the day after the celebration of the competitions where, after accepting to participate, the link to the NIQEC questionnaire with instructions was made available to the participants, who filled it in voluntarily, telematically, and anonymously. 

### 2.5. Statistical Analysis

All the variables collected were analyzed by means of descriptive statistics, obtaining the mean and standard deviation, as are shown in the tables. In order to analyze the differences in the intakes before, during, and after the sporting event, depending on the sport performed, an analysis of variance (ANOVA) was performed. In addition, a bivariate correlation was performed between the final time of the athletes and the intakes performed both during and before the event. The association between these variables was measured as follows: r < 0.3, low association; r = 0.3–0.5, moderate association; and r > 0.5, high association. As for the categorical variables, the chi-square test (X2) was performed with the aim of analyzing the frequency of gastrointestinal problems during the sporting event of the total sample and according to the sport they performed. The minimum level of statistical significance was set at *p* < 0.05. Data were analyzed using the Statistical Package for the Social Sciences (SPSS) version 25.0 (IBM, Armonk, NY, USA).

## 3. Results

According to the sport practiced by the athletes, Table 2 shows that the analysis of variance did not show significant differences in any of the variables analyzed (F = 0.006–0.611; *p* = 0.938–0.439).

Analyzing the intakes during the competitions, as shown in Table 3, which shows the total intakes, the analysis of variance only found significant differences for the consumption of sodium (F = 8.617; *p* = 0.005) and fluids (F = 13.099; *p* = 0.001), with the consumption of mountain runners being higher in both cases. However, as shown in Table 4, which shows the hourly intakes, the analysis of variance showed no significant differences in any of the variables analyzed (F = 0.001–2.764; *p* = 0.980–0.104).

As for after the event intakes, Table 5 shows that the analysis of variance only found significant differences for caffeine consumption (F = 7.106; *p* = 0.011), as mountain runners did not consume caffeine after exercise.

As presented in Table 6, and on the one hand, regarding the intakes performed during the event, energy/hour (r = −0.346; *p* = 0.025) and HC/hour (r = −0.380; *p* = 0.013) showed a moderate negative correlation with the final time. This means that the higher the energy and carbohydrate intake per hour, the lower the final time, or the other way around. On the other hand, with respect to the intakes before the event, only a moderate positive correlation was found between fat consumption per kilogram of weight and final time (r = 0.348; *p* = 0.024), indicating, in this case, that the higher the fat consumption, the longer the final time, or the other way around. 

Table 7 shows the frequencies of gastrointestinal problems in the total sample and according to the sport they were performing, with 61.9% of the cases showing some type of gastrointestinal discomfort. On the one hand, significant differences were only observed in the urge to defecate (X2 = 3.948; *p* = 0.047), where mountain runners had a higher frequency. On the other hand, and regarding the most frequent gastrointestinal problems, for the total sample and for the triathletes, the most frequent were belching (50.0% and 43.8%, respectively), gas (45.2% and 43. 8%, respectively), and stomach pain (45.2% and 40.6%, respectively), and for the mountain runners, the most frequent were belching (70.0%), reflux (60.0%), nausea (60.0%), stomach pain (60.0%), and the urge to defecate (70.0%). 

## 4. Discussion

This study measured the nutritional and fluid intakes and gastrointestinal discomfort in middle-distance triathlon runners and mountain runners with the use of a NIQEC questionnaire in a population of 42 athletes in total (*n* = 42). Among the main findings of this study is the appearance of significant differences in the total fluid and sodium consumption during the test, being higher in mountain runners. However, no significant differences appeared between the intakes of the different athletes neither the hour before nor during the test, calculated for each hour of competition, nor in the hour after, except in the case of caffeine which was mostly consumed by triathletes. In addition, a negative correlation was found between the energy consumed and the CHO consumed per hour during the event with the final time achieved in the test, and a positive correlation was found between the consumption of fat per kilogram of weight before the event and the final time achieved in the competition.

Concerning the consumption of the nutrients analyzed during the hour before the race, the dietary–nutritional recommendations indicate that, during the hour before, athletes should have an intake of up to 1 g/kg of CHO and 5–10 mL/kg (400–600 mL) of liquids [2,12]. These recommendations have been met in the athletes analyzed, with fluid intake being higher in the mountain runners than in the triathletes. This result tells us that the athletes were able to begin the competition with adequate liver glycogen reserves and a good state of hydration, being able to reduce the earlier onset of fatigue, endurance, or gut upset, although it is necessary to know what the nutritional intakes were during the 24 h−48 h prior [1,12,28]. Furthermore, the intake of proteins and fat was low, assuming a gastrointestinal benefit during the competition, despite the fact that there is no specific recommended amount of these macronutrients [1,7,12]. It is worth highlighting the low consumption of caffeine by the athletes, which is far from current recommendations for endurance sports [29]. This substance has been widely studied, and scientific studies suggest that an intake of 3–6 mg/kg during the hour before the event may have ergogenic effects, such as increases in endurance performance, increases in resistance capacity, or reductions in the perception of effort during exercise [30,31,32]. Athletes could be losing the usefulness of a safe ergogenic aids, with strong scientific evidence for their use in specific situations in sports, using evidence-based protocols [33]. Despite this, it is necessary for athletes to use and train previously with this substance due to its possible side effects [30].

Regarding CHO intakes during the competition, both groups are in an intake range below the general recommendation for endurance sports of 90 g/h of HC [9,10]. The consumption among the triathletes, at 46.56 ± 18.99 g/h CHO, is lower than previously described in other works in medium- and long-distance tests, where consumptions of approximately 62 and 71 g/h were reported [22], although it is relatively close to that found in cyclists in the same study (53 g/h). Regarding the mountain runners’ consumption of 34.88 ± 23.31 g/h CHO, this is a result very similar to that reported in previous studies, such as, for example, the work of Martinez et al. in a test with different distances, whose average intake was 32.2 g/h [24]. These data are also similar to the results described in marathon runners (35 g/h) [22], but are above other studies in mountain runners where they barely consumed 15 g/h [23]. However, considering other studies conducted specifically on ultramarathon races, the majority of the athletes were not able to reach the high CHO recommendations [34]. As described in previous results of the existing literature, most of these works report a consumption of between 20 and 40 g/h of CHOs for the vast majority of runners [35,36,37]. Therefore, it is recommended to work on this tolerance to reach higher intakes [13], through nutritional periodization and gastrointestinal training [13,14,28], since applying recommendations for shorter duration competitions could be an erroneous strategy for athletes [38]. This can help athletes achieve intakes of 60 g CHO/hour or more, which may improve their sports performance in these types of events [1,9,12,17].

Despite the importance of sodium when replacing electrolytes and preventing EAH, none of the groups reached the minimum of the recommendations established by the ACSM of 300–600 mg/h [19], this consumption being 269.95 ± 188.26 mg/h for the triathletes and 289.40 ± 165.54 mg/h for the mountain runners. The previous literature is far from the values reported in the case of the triathletes, where values of around 420 mg/h are reported [22]. In the case of the mountain runners, the reported values are in line with previous studies, at 142.6 mg/h [24] and 146.41 mg/h [23], although they are far from the 300–600 mg/h recommended by the ACSM [19]. Compliance with these recommendations regarding mineral intakes by athletes can help prevent hyponatremia. However, it is important to assess both mineral and fluid consumptions [9,12,14].

Respecting the liquid ingested during the competition, none of the groups met the recommendations of the International Society of Sports Nutrition (450–750 mL/h) [9] for ultra-distance tests, although both groups came close. To the lowest range recommended, the mountain runners came closer (447.15 ± 231.54 mL/h) than the triathletes (422.40 ± 175.51 mL/h). In the case of the consumptions reported by the triathletes, the values are lower than those previously reported, where consumptions were seen around 700 mL/h or higher [22]. As for the mountain runners, the consumptions reported in the present study are in accordance with the study by Martinez et al. [24], at about 459.2 mL/h, but are slightly higher than the results of Jiménez-Alfageme et al. [23], where 399.73 mL/h was reported. To achieve these previously described intakes, both triathletes and mountain runners include various foods, as well as supplements mainly included in the “Sport foods” subgroup belonging to Group A with maximum scientific evidence in the AIS classification [33], including mainly bars, gels, or sports drinks, as described in recent studies [39,40]. While these recommendations are a guide, each athlete should develop a personalized nutrition and hydration strategy during pre-competition training.

Regarding the consumption of the nutrients analyzed in the following hour, nutritional recommendations suggest an intake of CHO and proteins (0.8–1 g/kg of CHO and 0.2–0.4 g/kg of proteins) [2,7]. These values are met in the sample analyzed and can improve glycogen synthesis [2,7,12]. A nutrient that could have benefited the athletes and that can boost glycogen repletion by up to 66% more, is caffeine in doses of around 3 mg/kg [41], but as occurred in the analysis of the nutritional intakes in the hour before the competition, the intake of this nutrient is far from current nutritional recommendations [12,29].

In the matter of the prevalence of gastrointestinal problems suffered by athletes during the competitions, this stood at 61.9% of the total, and, depending on the specific problems considered, they were between 34.5% and 70%. These results support the idea that gastrointestinal problems in ultra-endurance sports are common [38], with an incidence of between 60 and 96% of severe problems of both the upper and lower gastrointestinal tract [34,42,43]. To minimize this incidence of problems during competitions, bowel training is necessary so that athletes can tolerate the recommended intake of carbohydrates, sodium, and fluids during training sessions or competition periods [13,14]. Other works indicate that this incidence is between 30 and 50% of endurance athletes [21], data that are relatively similar to those found in this work. However, these findings vary from the review conducted by Martínez-Sanz et al. in 2020, where, in more than 13,800 endurance athletes, the prevalence of gastrointestinal problems was only 9.4% [14]. 

### Limitations

This research has limitations that need to be discussed in order to improve its applicability to endurance athletes. The first limitation is the sample or low response rate. This was small, regarding the total number of participating athletes, and there is heterogeneity between the sex groups, but a significant sample in this type of population was used according to the statistical principles applied. Another limitation is that the information on consumption was collected in a self-reported and retrospective manner based on the athletes’ memory. This could lead to errors in the number and type of information reported (foods, supplements, liquids, or gastrointestinal problems). However, the questionnaire used was validated, unlike in other studies, where unvalidated or consensual questionnaires were employed by the research team. In addition, endurance athletes tend to worry about their diet and training, as their performance depends on it, and they are usually knowledgeable about their competition’s intake. This could make them better recall their intake compared to the general population. Therefore, the results of the current investigation may not be applicable to all participants of endurance competitions, as well as to other nations and cultures. This questionnaire has the strength of being able to estimate the type of fluid, food, or supplement consumed; the nutritional intake of kcal, macronutrients (CHO, lipids, and proteins), sodium, and caffeine; the incidence and causes of gastrointestinal discomfort and its relationship to food intake; and the compliance with dietary and nutritional recommendations for the hour before, during, and the hour after the competition.

## 5. Conclusions

The intake of energy, carbohydrates, water, sodium, and caffeine fell below current recommendations for both sports during the competition. However, these intakes met the recommendations for the previous hour of CHO and fluid intake, as well as the intake of CHO and proteins during the nutritional recovery phase (post-competition).

No significant differences existed in the energy, carbohydrate, fluid, sodium, and caffeine intakes between the triathletes and mountain runners. 

The gastrointestinal complaints showed a high prevalence in the endurance athletes, without differences between the mountain runners and triathletes.

Our data suggest the need to instruct endurance athletes to plan competitions at a dietary–nutritional level so that they can previously implement appropriate nutritional strategies. The advice of a nutrition professional, such as a dietitian–nutritionist, will help establish adequate nutritional periodization and gastrointestinal training to successfully compete in competitions and achieve the recommended nutritional requirements.

## Figures and Tables

**Table 1 nutrients-16-00189-t001:** Descriptive data of the participants according to the sport they practice.

Variables	Sports (Mean ± SD)	
	Triathlon (*n* = 32)	MR (*n* = 10)
Age (years)	41.84 ± 10.23	43.90 ± 5.62
Height (m)	1.75 ± 0.06	1.68 ± 0.06
Body mass (kg)	69.98 ± 7.13	61.92 ± 7.78
Body mass index–BMI (kg/m^2^)	22.82 ± 1.57	21.86 ± 1.49
Years of experience	8.31 ± 8.00	7.30 ± 3.37
Training session per week	6.06 ± 1.95	5.00 ± 1.33
Training hours per week	10.16 ± 3.18	10.30 ± 3.09

**Table 2 nutrients-16-00189-t002:** Descriptive data and ANOVA of the intakes per kilograms of weight of the participants before the competition, according to the sport they performed.

Variables	Sports (Mean ± SD)		ANOVA	
	Triathlon (*n* = 32)	MR (*n* = 10)	F	*p*
Energy/body mass (kcal/kg)	5.80 ± 5.97	6.13 ± 5.91	0.023	0.880
CHO/body mass (g/kg)	1.09 ± 1.28	1.00 ± 1.16	0.037	0.848
Proteins/body mass (g/kg)	0.15 ± 0.16	0.16 ± 0.16	0.026	0.873
Lipids/body mass (g/kg)	0.13 ± 0.18	0.16 ± 0.17	0.271	0.605
Sodium/body mass (mg/kg)	4.98 ± 6.18	6.80 ± 7.15	0.611	0.439
Caffeine/body mass (mg/kg)	0.28 ± 0.65	0.26 ± 0.35	0.006	0.938
Fluid/body mass (mL/kg)	9.83 ± 7.49	12.91 ± 20.27	0.532	0.470

Kcal: kilocalories; kg: kilograms; CHO: carbohydrates; g: grams; mg: milligrams; mL: milliliters.

**Table 3 nutrients-16-00189-t003:** Descriptive data and ANOVA of the total intakes of the participants during the sporting event, according to the sport performed.

Variables	Sports (Mean ± SD)		ANOVA	
	Triathlon (*n* = 32)	MR (*n* = 10)	F	*p*
Energy (kcal)	1100 ± 435	1364 ± 798	1.821	0.185
CHO (g)	205.10 ± 79.21	303.80 ± 187.31	1.525	0.224
Proteins (g)	8.05 ± 8.24	12.14 ± 7.98	1.903	0.175
Lipids (g)	7.35 ± 10.32	9.79 ± 13.35	0.368	0.548
Sodium (mg)	1437 ± 1047	2881 ± 2104	8.617	0.005
Caffeine (mg)	88.90 ± 149.10	179.27 ± 365.99	1.314	0.259
Fluid (mL)	2275 ± 1003	4211 ± 2495	13.099	0.001

Kcal: kilocalories; kg: kilograms; CHO: carbohydrates; g: grams; mg: milligrams; mL: milliliters.

**Table 4 nutrients-16-00189-t004:** Descriptive data and ANOVA of the hourly intakes of the participants during the sporting event, according to their sport.

Variables	Sports (Mean ± SD)		ANOVA	
	Triathlon (*n* = 32)	MR (*n* = 10)	F	*p*
Energy/hour (kcal/h)	205.10 ± 79.22	154.79 ± 96.91	2.764	0.104
CHO/hour (g/h)	46.56 ± 18.99	34.88 ± 23.31	2.586	0.116
Proteins/hour (g/h)	1.45 ± 1.41	1.23 ± 0.60	0.218	0.643
Lipids/hour (g/h)	1.28 ± 1.73	1.00 ± 1.25	0.218	0.643
Sodium/hour (mg/h)	269.95 ± 188.26	289.40 ± 165.54	0.086	0.771
Caffeine/hour (mg/h)	15.99 ± 25.62	15.75 ± 27.44	0.001	0.980
Fluid/hour (mL/h)	422.40 ± 175.51	447.15 ± 231.54	0.130	0.720

Kcal: kilocalories; kg: kilograms; CHO: carbohydrates; g: grams; mg: milligrams; mL: milliliters.

**Table 5 nutrients-16-00189-t005:** Descriptive data and ANOVA of the intakes per kilograms of weight of the participants after the sporting event, according to the sport they performed.

Variables	Sports (Mean ± SN)		ANOVA	
	Triathlon (*n* = 32)	MR (*n* = 10)	F	*p*
Energy/body mass (kcal/kg)	7.79 ± 6.28	9.09 ± 7.25	0.304	0.584
CHO/body mass (g/kg)	1.03 ± 0.75	0.98 ± 0.73	0.037	0.849
Proteins/body mass (g/kg)	0.23 ± 0.25	0.40 ± 0.43	2.452	0.125
Lipids/body mass (g/kg)	0.29 ± 0.32	0.37 ± 0.36	0.461	0.501
Sodium/body mass (mg/kg)	12.08 ± 14.24	17.10 ± 19.29	0.799	0.377
Caffeine/body mass (mg/kg)	0.28 ± 0.33	0.00 ± 0.00	7.106	0.011
Fluid/body mass (mL/kg)	10.72 ± 6.02	10.39 ± 7.25	0.020	0.887

Kcal: kilocalories; kg: kilograms; CHO: carbohydrates; g: grams; mg: milligrams; mL: milliliters.

**Table 6 nutrients-16-00189-t006:** Bivariate correlation between the final time of the athletes in their corresponding events and the intakes before and during the event.

During the Competition		Before the Competition	
Intakes	Final Time	Intakes	Final Time
Energy/hour (kcal/h)	r = −0.346; *p* = 0.025	Energy/body mass (kcal/kg)	r = 0.228; *p* = 0.146
CHO/hour (g/h)	r = −0.380; *p* = 0.013	CHO/body mass (g/kg)	r = 0.122; *p* = 0.440
Proteins/hour (g/h)	r = 0.059; *p* = 0.710	Proteins/body mass (g/kg)	r = 0.286; *p* = 0.067
Lipids/hour (g/h)	r = 0.062; *p* = 0.696	Lipids/body mass (g/kg)	r = 0.348; *p* = 0.024
Sodium/hour (mg/h)	r = 0.092; *p* = 0.561	Sodium/body mass (mg/kg)	r = 0.293; *p* = 0.059
Caffeine/hour (mg/h)	r = 0.164; *p* = 0.299	Caffeine/body mass (mg/kg)	r = 0.044; *p* = 0.784
Fluid/hour (mL/h)	r = 0.036; *p* = 0.821	Fluid/body mass (mL/kg)	r = 0.239; *p* = 0.128

Kcal: kilocalories; kg: kilograms; CHO: carbohydrates; g: grams; mg: milligrams; mL: milliliters.

**Table 7 nutrients-16-00189-t007:** Frequency (%) of gastrointestinal problems suffered by the total number of athletes and according to their sport.

Gastrointestinal Complaints	Total (%)	Triathlon (%)	MR (%)	Statistics Values	
				X2	*p*
Belching	50.0	43.8	70.0	2.100	0.147
Heartburn	40.5	37.5	50.0	0.494	0.482
Reflux	42.9	37.5	60.0	1.575	0.209
Bloating	38.1	34.4	50.0	0.789	0.374
Gas	45.2	43.8	50.0	0.120	0.729
Dizziness	38.1	34.4	50.0	0.789	0.374
Nausea	40.5	34.4	60.0	2.077	0.150
Vomiting	38.1	34.4	50.0	0.789	0.374
Flatus	40.5	37.5	50.0	0.494	0.482
Stomach pain	45.2	40.6	60.0	1.155	0.283
Intestinal pain left	38.1	34.4	50.0	0.789	0.374
Intestinal pain right	40.5	37.5	50.0	0.494	0.482
Urge to defecate	42.9	34.4	70.0	3.948	0.047
Def. normal consistency	38.1	34.4	50.0	0.789	0.374
Abnormal def. loose stools	38.1	34.4	50.0	0.789	0.374
Def. diarrhea	38.1	34.4	50.0	0.789	0.374
Def. bloody stool	38.1	34.4	50.0	0.789	0.374

Def.: Defecation.

## Data Availability

The data from this study are available on request from the corresponding author.

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
