# Peer review of "Are the Dietary–Nutritional Recommendations Met? Analysis of Intake in Endurance Competitions"

_nutrients, 2024, doi:10.3390/nu16020189_

Round 1

Reviewer 1 Report

Comments and Suggestions for Authors

Dear

The present manuscript brings the theme relating to the dietary intake and compliance with nutritional recommendations of athletes in endurance competitions. Although this theme is highly relevant for athletes, the concepts and recommendations have been in literature since the early 80’s. The authors were not able to bring new information or solutions to apply in athletes to minimize either gastrointestinal problems or increase adherence to already pre-established recommendations.

Author Response

The present manuscript brings the theme relating to the dietary intake and compliance with nutritional recommendations of athletes in endurance competitions. Although this theme is highly relevant for athletes, the concepts and recommendations have been in literature since the early 80’s. The authors were not able to bring new information or solutions to apply in athletes to minimize either gastrointestinal problems or increase adherence to already pre-established recommendations.

Response of the authors: we welcome the reviewer's comments. It is true that dietary-nutritional recommendations were developed decades ago, but they have evolved over time. Research on dietary-nutritional intake in various endurance sports is very limited and despite the information available, athletes generally still do not comply with these recommendations.

In addition, a phrase has been included in the discussion to help athletes suffer a lower incidence of gastrointestinal problems in their training and sporting events.

Reviewer 2 Report

Comments and Suggestions for Authors

The paper presented for review: „Are the Dietary-Nutritional Recommendations Met? Analysis of Intake in Endurance Competitions” is a observational and cross-sectional studywork with a classic layout. The authors analyze the dietary intake and compliance with nutritional recommendations of athletes in two endurance competitions, as well as to assess the incidence of gastrointestinal problems.

The paper needs minor revision:

The introduction is well written, in which the Authors describe in detail nutritional recommendations for endurance athletes.  I have a few suggestions:

Line 52: please remove „From the energetic point of view”.

Table 1: please use „body mass” instead of „weight”; use superscript in BMI units (kg/m2)

Did the athletes or their legal guardians give written consent to participate in the research?

Results

I recommend to rewrite this section. I suggest stated in the Materials and Methods section, that data in tables are shown as mean and standard deviations and  describe the results in a different way, e.g.: „As presented in Table 2, analysis of variance showed no significant differences in any of the variables analyzed (F = 0.006 - 0.611; p = 0.938 - 0.439).

 Tables 2, 5 and 6: please use „body mass” instead of „weight”

Discussion

The discussion is also well written, however I have a few suggestions.

Lines 209 - 314: please write „fluids” instead of „water” because during sports events athletes consume not only water but also other fluids.

In the summary lacks the statement that the recommendations are a guide, but each athlete should develop a nutrition and hydration strategy for ultra competitions during training and previous competitions.

Author Response

The paper presented for review: „Are the Dietary-Nutritional Recommendations Met? Analysis of Intake in Endurance Competitions” is a observational and cross-sectional studywork with a classic layout. The authors analyze the dietary intake and compliance with nutritional recommendations of athletes in two endurance competitions, as well as to assess the incidence of gastrointestinal problems.

 The paper needs minor revision:

 The introduction is well written, in which the Authors describe in detail nutritional recommendations for endurance athletes.  I have a few suggestions:

Line 52: please remove „From the energetic point of view”.

Table 1: please use „body mass” instead of „weight”; use superscript in BMI units (kg/m2)

Response of the authors: we thank the reviewer's comments, the suggested changes have been made to improve the manuscript.

Did the athletes or their legal guardians give written consent to participate in the research?

Response of the authors: Yes, the did as is indicated in the paragraph “Informed Consent Statement”: Informed consent was obtained from all subjects involved in the study. (Lines 343-344).

Results

I recommend to rewrite this section. I suggest stated in the Materials and Methods section, that data in tables are shown as mean and standard deviations and  describe the results in a different way, e.g.: „As presented in Table 2, analysis of variance showed no significant differences in any of the variables analyzed (F = 0.006 - 0.611; p = 0.938 - 0.439).

Response of the authors: We appreciate the reviewer comments. It has been indicated in the statistical analysis section under Materials and methods that the data in the tables are shown as means and standard deviations. The results have been rewritten as suggested by the reviewer.

 Tables 2, 5 and 6: please use „body mass” instead of „weight”

Response of the authors: we thank the reviewer's comments, the suggested changes have been made to improve the manuscript.

Discussion

The discussion is also well written, however I have a few suggestions.

Lines 209 - 314: please write „fluids” instead of „water” because during sports events athletes consume not only water but also other fluids.

Response of the authors: we thank the reviewer's comments, the suggested changes have been made to improve the manuscript.

In the summary lacks the statement that the recommendations are a guide, but each athlete should develop a nutrition and hydration strategy for ultra competitions during training and previous competitions.

Response of the authors: we thank the reviewer's comments, the information suggested by the reviewer has been included in the discussion section.

Reviewer 3 Report

Comments and Suggestions for Authors

Thank you for submitting the manuscript "Are the Dietary-Nutritional Recommendations Met? Analysis of Intake in Endurance Competitions" to Nutrients. The manuscript evaluated athletes in two types of competition, indicating the relationship between food and drink intake and gastrointestinal discomfort. Overall, the manuscript is clear and concise, and the experiments appear to have been well conducted. However, I have some considerations:

- The authors themselves cite other works that have already carried out the same type of evaluation that was carried out in this research. This left a gap, in which this work is different from the others, that is, what is the justification for carrying out and reporting more research in this sense since there already seems to be a consensus that athletes do not meet the recommendations during competitions? This needs to be clear in the introduction.

- Line#131: no supplementary files were provided.

- Table 2, 3, 4, and 5 should show the recommended average and how much percentage of this average was achieved for a more dynamic reading of the results.

- It is necessary to try to combine the tables as I found the results very fragmented which greatly increased the number of elements (tables/figures) in the manuscript.

Author Response

Thank you for submitting the manuscript "Are the Dietary-Nutritional Recommendations Met? Analysis of Intake in Endurance Competitions" to Nutrients. The manuscript evaluated athletes in two types of competition, indicating the relationship between food and drink intake and gastrointestinal discomfort. Overall, the manuscript is clear and concise, and the experiments appear to have been well conducted. However, I have some considerations:

- The authors themselves cite other works that have already carried out the same type of evaluation that was carried out in this research. This left a gap, in which this work is different from the others, that is, what is the justification for carrying out and reporting more research in this sense since there already seems to be a consensus that athletes do not meet the recommendations during competitions? This needs to be clear in the introduction.

Response of the authors: we welcome the reviewer's comments. A more detailed explanation of the justification for the work performed has been included in the last paragraph of the introduction.

- Line#131: no supplementary files were provided.

Response of the authors: As indicated in the supplementary material section, this can be found in the following link: https://www.mdpi.com/article/10.3390/nu15081969/s1 which corresponds to the specific article where the questionnaire used was developed.

- Table 2, 3, 4, and 5 should show the recommended average and how much percentage of this average was achieved for a more dynamic reading of the results.

Response of the authors: we thank the reviewer's comments. Recommendations have not been included in the tables, as some of them are not fully researched or sufficiently detailed in the literature. In addition, the most relevant recommendations are cited throughout the text and the authors prefer not to duplicate such information.

The recommendations have not been included in the tables, as some recommendations depend on race pace, weather, intensity, etc. and there can be large individual variability. Wide ranges can be observed and it is difficult to classify the athletes evaluated due to the sample size (limitation indicated in the corresponding section). However, a paragraph has been added to indicate the average hourly consumption of triathletes and mountain runners in relation to the recommendations.

- It is necessary to try to combine the tables as I found the results very fragmented which greatly increased the number of elements (tables/figures) in the manuscript.

Response of the authors: we appreciate the reviewer’s comments. Each table represents a moment in the race: 1 hour before, during and 1 hour after. And as all the recommendations are also given according to the time of the race, the tables are in that order. In addition, putting them together would result in excessively large tables that would be more difficult for the reader to understand.

Round 2

Reviewer 1 Report

Comments and Suggestions for Authors

Dear,
No further suggestion